# Prevalence and risk factors for hepatitis C virus infection in an informal settlement in Karachi, Pakistan

**Munazza Mansoor**[1]*, **William A. de Glanville**[2], **Ridwa Alam**[1], **Khawar Aslam**[2], **Mubashir Ahmed**[2], **Petros Isaakidis**[3], **Aneeta Pasha**[1]

1 Interactive Research and Development (IRD), Karachi, Pakistan, 2 Médecins Sans Frontières (MSF), Brussels, Belgium, 3 Southern Africa Medical Unit, Médecins Sans Frontières, Cape Town, South Africa

* munazza.mansoor@ird.global

**Data Availability Statement:** The data that support the findings of this study are openly available as supplementary information provided as part of the manuscript.

## Abstract

The burden of hepatitis C virus (HCV) infection in Pakistan is amongst the highest in the world. People living in slums are likely to be at high risk of infection. Here, we describe the results of a cross-sectional survey conducted in March 2022 that aimed to quantify the prevalence of HCV infection in Machar Colony, one of the largest and oldest slum settlements in Karachi. Risk factors for HCV seropositivity were identified using multi-level logistic regression. We recruited 1,303 individuals in a random selection of 441 households from Machar Colony. The survey-adjusted HCV-seroprevalence was 13.5% (95% Confidence Interval (CI) 11.1–15.8) and survey-adjusted viraemic prevalence was 4.1% (95% CI 3.1–5.4) with a viraemic ratio of 32% (95% CI 24.3–40.5). Of 162 seropositive people, 71 (44%) reported receiving previous treatment for chronic hepatitis C. The odds of HCV seropositivity were found to increase with each additional reported therapeutic injection in the past 12 months (OR = 1.07 (95% Credible Interval (CrI) 1.00–1.13)). We found weaker evidence for a positive association between HCV seropositivity and a reported history of receiving a blood transfusion (OR = 1.72 (95% CrI 0.90–3.21)). The seroprevalence was more than double the previously reported seroprevalence in Sindh Province. The overall proportion of seropositive people that were viraemic was lower than expected. This may reflect the long-term impacts of a non-governmental clinic providing free of cost and easily accessible hepatitis C diagnosis and treatment to the population since 2015. Reuse of needles and syringes is likely to be an important driver of HCV transmission in this setting. Future public health interventions should address the expected risks associated with iatrogenic HCV transmission in this community.

## 1. Introduction

An estimated 57 million people are infected with the hepatitis C virus (HCV) worldwide [1]. Infection with HCV can range in severity from a mild illness with viral clearance to chronic hepatitis C (CHC) that may lead to liver failure and death [2]. There are an estimated 411,000

**Funding:** This Project was funded by Foundation for Innovative New Diagnostics. The funders had no role in the study design, data collection and analysis, decision to publish, or preparation of the manuscript.

**Competing interests:** The authors have declared that no competing interests exist.

annual HCV-attributable deaths globally, with most of these resulting from liver cirrhosis or hepatocellular carcinoma that can manifest decades after infection [2–5].

Pakistan has the second highest HCV infection prevalence after Egypt [6]. One in every twenty individuals in the country is estimated to have been infected with the virus [7, 8]. The main route of HCV transmission in Pakistan is thought to be through healthcare-related exposures [7]. Contributory factors to this iatrogenic transmission include poor sterilization of medical equipment and the re-use of needles and syringes for therapeutic injections, as well as limited screening of, and transfusion with, contaminated blood products [7]. Community-level exposures such as barbering, tattooing, and piercing are also thought to be important [2, 9, 10]. Intravenous drug use is increasing in Pakistan [11] and is also likely to play an increasingly important role in HCV transmission. A recent meta-analysis estimated an HCV sero-prevalence of 54% among people who inject drugs (PWID) in Pakistan [12].

Hepatitis C is a curable disease, with direct acting antivirals (DAAs) capable of achieving cure rates of above 90% [13]. The price of these drugs has decreased substantially over the past decade, making the global elimination of HCV as a public health problem a realistic target [14]. The WHO has created a timeline for the global elimination of the virus by 2030 [15]. In line with this, the Government of Pakistan has developed the National Hepatitis Strategic Framework for Pakistan [8]. By 2030, Pakistan aims to reduce the incidence of hepatitis C by 90%, reduce mortality due to the disease by 65%, to diagnose 90% of hepatitis C cases, and to provide treatment to 80% of persons eligible for treatment [8].

Limited data exist, but people living in slums are likely to be at particularly high risk for HCV infection. High levels of poverty and limited access to formal medical care within these settings is likely to drive the use of informal health care providers [16, 17]. Informal health care providers such as untrained dispensers, homeopaths, and hakims, commonly give injections and perform invasive procedures but often have little or no medical training or knowledge of infection prevention and control [18]. Where formal health provision is available in slum areas, it can be of poor quality [19], which may also increase the risk of iatrogenic infection. Limited access to formal care also suggests people with chronic hepatitis C are likely to remain untreated for long periods and can therefore act as a source of HCV infection for others in the community. More than 50% of the population of major cities in Pakistan live in slums and squatter settlements [20]. Effective HCV screening and treatment in these high-risk areas is therefore likely to be critical for achieving Pakistan's 2030 elimination target, as well as for reducing the burden of hepatitis C disease in these marginalised communities.

Interactive Research and Development (IRD) and Médecins Sans Frontières (MSF) collaborated to conduct a prevalence survey within the informal settlement of Machar Colony to understand the prevalence within this community. In this study, we aimed to estimate the sero- and viraemic prevalence of HCV infection in Machar Colony, one of the largest and oldest slum settlements in Karachi. We also aimed to identify risk factors for HCV infection that could be used to target future public health interventions in this marginalised setting.

## 2. Methods

### 2.1. Ethical approval

This research was approved by Interactive Research and Development Institutional Review Board (IRD_IRB_2021_09_001), the Médecins sans Frontières Ethics Review Board (2166), and the Pakistan National Bioethics Committee Research Ethics Committee (No.4-87/NBC-749/22/1549). All participants aged 18 and above provided written informed consent. All participants aged between 12 and 18 provided written assent and their parent or legal guardian provided written consent.

## 2.2. Study setting

The study was conducted in Karachi, the largest metropolitan city of Pakistan, with a population of over 16 million [21]. The survey was conducted in a union council called Machar Colony located within Kemari Town, Kemari District, Karachi. The name 'Machar' is derived from 'Machera' (fisherman) with the primary source of income within the colony being fishing and shrimp processing. It is an unplanned settlement located near the Port of Karachi and Lyari. Machar Colony comprises an area of approximately 1.8 square kilometers and is home to an estimated 150,434 residents, constituting a population that includes many migrants and their descendants from across Pakistan as well as from Afghanistan, Bangladesh and Myanmar [22].

Since 2015, two Médecins sans Frontières (MSF) clinics have consecutively offered free of cost hepatitis C screening and chronic hepatitis C (CHC) treatment using DAA therapy to the population of Machar Colony. Between 2015 and 2017, testing was offered to patients meeting risk-based screening criteria for hepatitis C within an MSF managed primary health care clinic located centrally within Machar Colony. In 2017, a vertical clinic focusing on HCV diagnosis and management was established in Machar Colony. This model of care was intended to provide hepatitis C-focused care from screening to cure for the population of Machar Colony and surrounding areas [23]. Health promotion messaging related to hypothesized drivers of HCV infection in the community was also initiated, with community members encouraged to attend the clinic for screening during these outreach activities. In addition, between August 2018 and December 2021, MSF conducted 22 community screening "camps" at sites throughout Machar Colony, with linkage to care at the MSF clinic for those found to be screening test positive.

## 2.3. Study design

We used a cross-sectional design to estimate the population-level sero- and viraemic prevalence of HCV infection in Machar Colony. Households were the primary sampling unit with all household members aged 12 years and above eligible for inclusion. Eligibility included any person who had stayed in the household the previous night.

**Sampling framework.**   Based on estimates from MSF screening camp data, we expected the seroprevalence in this setting to be around 15%. We estimated we would require a sample size of 780 people to estimate a prevalence of 15% with 2.5% error at the 95% confidence interval based on simple random sampling [24]. To adjust this sample size for potential clustering of HCV infection at the household level [25], we used an estimated intra-cluster correlation co-efficient (ICC) of 33%. Assuming we would recruit an average of three individuals 12 years or older per participating household (based on an average total household number in Pakistan of 5.8 from the 2017 National Pakistan Census) [22], we multiplied our sample size by a design effect of 1.66 [24]. This resulted in a total target sample size of 1,300 people, expected to be found in 434 households.

No sampling frame for households existed in this setting. We therefore split Machar Colony into 188 grid cells (Fig 1). A target number of households were then selected from each grid cell for recruitment. For the 50 (27%) incomplete grid cells (i.e., those around the edge of Machar Colony), the household sample (up to a maximum of three) was selected proportional to the area of the grid cell. For the 138 (73%) complete grid cells, approximately 40% were randomly allocated to a household sample of two and the remaining 60% to a household sample of three. Using this system, a target of one or more households was selected in 183 (97%) of the 188 grid cells.

For household selection, the target number of households for each grid cell were generated as random points using QGIS version 3.22.1 (https://qgis.org). A fixed number of 'backup'

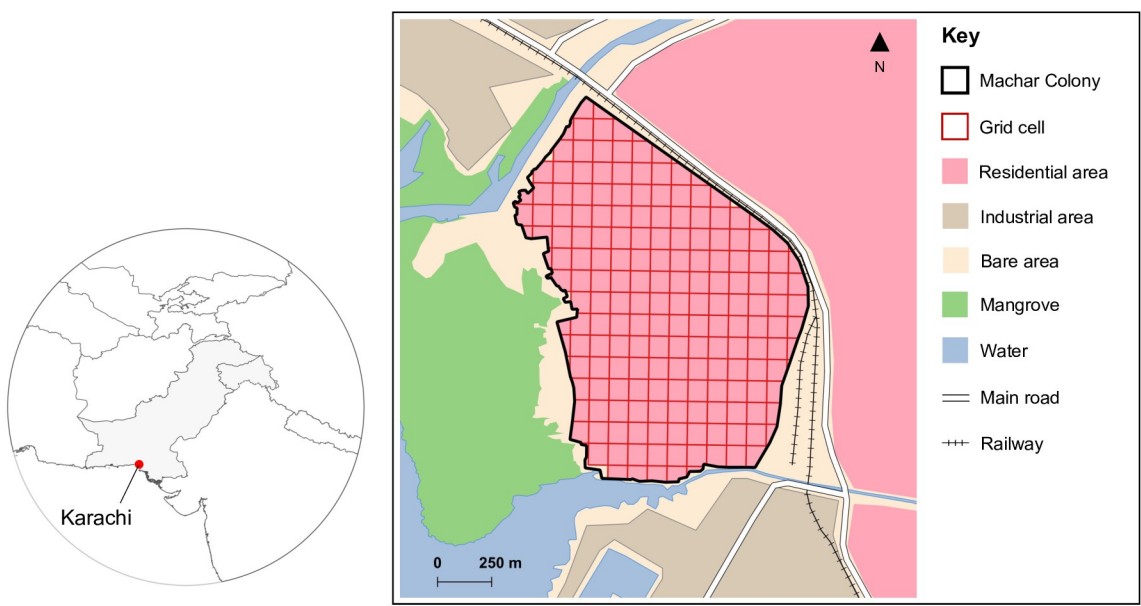

**Fig 1. Map of study site in Machar Colony, Karachi.** Position of Karachi in Pakistan also shown. Maps generated in QGIS version 3.22.3. Shapefiles from GADM (https://gadm.org/data.html) and generated by authors.

points was also generated. The study team then utilized the GPS on smartphones to get as close as possible to each random point while remaining outside buildings. The housing unit closest to the point was selected for inclusion. If the selected building had multiple levels containing multiple housing units, a random number generator was used to select the level and then a household on that level for recruitment. In the case that households refused to be involved in the study, replacement households in the grid cell were selected using the "back-up" random point.

If no one was present in the randomly selected household at the time of first visit, or the main household decision-maker was not present, it was revisited at least once the following day before a replacement back-up household was selected for recruitment. Recruitment and data collection took place during daylight hours between 8 a.m. and 5 p.m. Data collection included weekends to maximise recruitment, with households revisited during the weekend where household members were absent during the week.

## 2.4. Diagnostic testing

Study participants were tested in their homes for antibodies to HCV using the WHO pre-qualified SD Bioline rapid diagnostic test (RDT) (Abbott Diagnostics Korea Inc, South Korea) from a finger prick sample. Any individual with a negative RDT was given post-test counselling on ways to reduce their risk of HCV infection. Any person that was RDT positive was referred to the MSF hepatitis C clinic in Machar Colony for confirmatory PCR-based testing. At the MSF clinic, individuals were tested for current HCV infection using a GeneXpert quantitative PCR (Cepheid, Sunnyvale, CA, USA).

Individuals found to have HCV infection were managed as patients following standard MSF clinical protocols. These patients had consultations with medical doctors at the MSF hepatitis C clinic. This included detailed history taking and examination. Patients were provided with a combination therapy of Sofosbuvir (SOF)/Daclatasvir (DAC) 400mg/60mg/day (which

needs to be taken orally for 12 weeks) as the primary treatment for all aged 12 years or above. This treatment is considered to be very potent and well tolerated, including in patients with advanced liver disease and has a pan-genotypic profile. This regimen is recommended by the WHO and Pakistan National HCV guidelines for the treatment of adults with CHC, with and without compensated cirrhosis [26]. No charge was made for any diagnostic testing or treatment. Individuals that had not attended the MSF clinic within one week of a positive RDT were telephoned or, in the absence of a telephone or where no answer was received, visited in person by the IRD team to ensure that they had understood the need for confirmatory testing and to encourage attendance of the MSF clinic.

## 2.5. Data collection

Participating household members were administered an individual questionnaire with close-ended questions focusing on behaviours and exposures that have been shown to be important as risk factors for HCV infection in other settings in Pakistan [9, 10, 27, 28]. All individuals tested with the RDT were asked about their previous HCV testing history and if they had ever received treatment for HCV from a doctor. All data were collected on digital devices via Open Data Kit (ODK). Questionnaires used are included in S3 File. All data collection from households was conducted by trained staff from IRD.

## 2.6. Data analysis

**2.6.1. Summary statistics.** We estimated the seroprevalence (proportion of RDT positive people), viraemic prevalence (proportion of PCR positive people), and the proportion of people reporting ever having tested for HCV. For each estimate, we incorporated a clustering variable and sampling weights at the household level to derive survey-adjusted estimates and associated 95% confidence intervals. Although our target was to screen all eligible people in the household, we expected that not all would agree to be sampled or would be available for sampling. Sampling weights were therefore estimated as the inverse of the fraction of people aged 12 and above sampled in the household and the number of people 12 and above living in the household. Adjustment of these estimates for survey design was implemented using the *survey* package in R version 4.1.1 [29]. We estimated the viraemic ratio as the proportion of seropositive people that were PCR positive. Exact binomial 95% confidence intervals were derived for this estimate using the *binom* package in R.

**2.6.2. Predictors of HCV exposure.** The relationship between individual-level characteristics derived from the questionnaire survey and the odds of HCV seropositivity were evaluated using logistic regression.

Individual-level variables for inclusion in the analysis were selected based on *a priori* knowledge about expected drivers of HCV transmission in Machar Colony. The main driver of HCV infection in this setting is thought to be unsafe medical injections. Reported number of medical injections received over the past 12 months was therefore included as a primary variable of interest. We have also heard anecdotal reports of potentially unsafe practices in the fisheries sector in Machar Colony, such as fishermen and fish factory workers sharing knives for gutting fish and deveining shrimp. We therefore included a binary indicator for those people who reported being either a fisherman or working in fish processing. Individual-level characteristics found to be important in other settings, including a history of blood transfusion and ever using a dentist were also included. On the expectation that knowledge of HCV transmission may influence risk behaviour, we included a single binary indicator for "HCV awareness" for any individual reporting that the risk of HCV infection could be reduced by giving birth in a hospital, not sharing toothbrushes or razors, through using single use syringes and avoiding

unsafe injections, safe sex, avoiding unsafe blood transfusions, or avoiding contact with potentially contaminated body fluids.

We first evaluated the relationship between HCV seropositivity and each hypothesized predictor using univariable logistic regression. All predictors of HCV seropositivity, as well as age and sex, were then included in a full multivariable model without model selection.

**2.6.3 Predictors of HCV viraemia.** We also assessed the relationship between HCV viraemia and age and sex among people that were HCV seropositive using logistic regression. A binary variable for whether the seropositive individual reported ever receiving treatment for HCV was also included. We first evaluated the relationship between HCV viraemia and age, sex, and previous treatment using univariable logistic regression. All predictors were then included in a full multivariable model without model selection.

**2.6.4. Logistic regression model specification.** Logistic regression models were formulated within a Bayesian framework using Just Another Gibbs Sampler (JAGS) software. To explore partitioning of variance in HCV seropositivity at the household and grid level, we compared three 'null' (intercept only) models containing either a random effect at the household, grid, or household and grid levels. The random effects structure with the lowest predictive error based on the deviance information criterion (DIC) was then used for univariable and the multivariable logistic regressions of HCV seropositivity. No random effects were included in the model for HCV viraemia.

For the multivariable logistic regression, we derived the posterior probability that the estimated odds ratio was less than one for hypothesized risk factors and greater than one for hypothesized protective effects [30]. Risk factors (with an expected odds ratio greater than one) were an increasing number of therapeutic injections received in the past 12 months, work in the fishing sector, reported history of receiving a transfusion, and reported use of a dentist in an individual's lifetime. The hypothesized protective effect (with an expected odds ratio less than one) was our indicator of HCV awareness. Age and sex were included in the multivariable model to provide control for potential confounding of the relationship between these hypothesised risk factors and protective effects and HCV seropositivity. Anon-linear relationship between HCV seropositivity and age, has been widely observed [31], increasing initially in young age before plateauing in middle and old age. We therefore included a quadratic term with age.

Logistic regressions were performed in JAGS via the R package, *R2jags* [32] in R 4.1.1. Where missingness was present in the predictor variables included in the multivariable model of HCV seropositivity missing values were estimated by specifying additional logistic regression models for binary categorical variables and a negative binomial regression for the number of reported injections received in the past 12 months. Age and sex were included as the only predictors in these additional regressions. No random effects were included. These additional regressions allowed the most likely missing values to be estimated for each variable based on its relationship with age and sex and the frequency of the variable in the population [33]. Missing values present in the HCV viraemia response (i.e., because seropositive people did not attend the clinic for PCR-based testing), were estimated directly from the multivariable model of HCV viraemia. To derive survey-adjusted estimates of HCV viraemia prevalence, we drew 1000 samples from the posterior distribution for HCV viraemia (with each draw comprising a prediction of the HCV viraemia status for those people with missing PCR results and the observed HCV viraemia status for those people attending the clinic) and derived the average prevalence and 95% confidence interval following the survey adjustment approach described in Section 2.6.1. The R code to implement the multivariable logistic regressions, including missing value imputation, is given in S4 File.

Convergence for all parameter estimates in the logistic regression models was assessed by visual examination of three MCMC chains after a minimum burn-in of 50,000 and at least 100,000 iterations. Weakly informative normal priors were used for fixed and random effects. Precision for random effects in the full logistic regression of seropositivity was defined using a wide uniform hyperprior [34].

**2.6.5. Spatial analysis.** The highest-level residual log odds (i.e., household or grid, depending on the random effects structure with the lowest DIC) were extracted from the logistic regression model of HCV seropositivity without predictors (the 'null' model) and from the model with all predictors (the 'full' multivariable model). The residual log odds of seropositivity from both the null and full models were then evaluated for evidence of residual spatial autocorrelation (RSA) using the Moran's I statistic. The Moran's I statistic was estimated using the *sdpep* package [35] in R. The Moran's I statistic from the null and full multivariable models were compared to assess if the included predictors explained any RSA observed. Clustering of high and low residual log odds of seropositivity from the null model was also characterised using a circular spatial scan statistic in SatScan version 10 (https://www.satscan.org/). Weighting for a weighted normal model in SatScan was provided by the number of people tested for HCV antibodies per grid cell [36, 37]. Grid cell centroids extracted in QGIS version 3.22.1 provided spatial information.

# 4. Results

## 4.1. Summary statistics

A total of 1,303 people were tested for HCV antibodies. Of these, 162 (12.4%) were seropositive. The survey-adjusted seroprevalence was 13.5% (95% Confidence Interval (CI) 11.1–15.8). A total of 135 (83%) of all RDT positive people had presented to the MSF clinic for PCR-based confirmatory testing by the end of August 2022. Of these, 43 were PCR positive, representing a viraemic ratio of 32% (95% CI 24.3–40.5). The survey-adjusted viraemic prevalence was 4.1% (95% CI 3.1–5.4).

Of the 1273 people (98%) providing an answer to the question "have you ever been tested for hepatitis C", 300 (24%) reported "yes". The survey-adjusted prevalence of reported previous testing was 26.9% (95% CI 23.4–30.4). Of these 300 people, 232 (77%) reported that testing had been performed in Machar Colony. Of the 162 people with a positive RDT, 71 (44%) reported ever having received treatment for chronic hepatitis C. Of these 71, 68 (96%) reported having completed treatment. Three (4%) of the 71 RDT positive individuals reporting previous treatment were PCR positive. All three reported having completed treatment but we have no details on the date of treatment, treatment duration or therapy received.

There was a clear increase in seropositivity by age with a plateau effect apparent beyond around 45 years (Fig 2).

Table 1 gives the characteristics of people included in this study and the relationship of these characteristics with HCV seropositivity. There was up to a maximum of 6% missingness in categorical variables. Table 1 shows the number of available responses for each variable. Of the 566 (46%) people reporting receiving a therapeutic injection in the past 12 months, 198 (35%) were unable to recall the number of injections they had received. For those people that did give an answer, the mean number was 1.5 (median = 0, range 0 to 30). For people reporting any injection in the past 12 months, the mean number of injections received was 3.8 (median = 3.0, range 1–30). The median age was 28 (range 12 and 90), equivalent to the median age of 29 reported among people above 12 in urban Sindh in the 2017 census [22]. There was no missingness in age data. Three individuals reported being transgender. Transgender are recognized in Pakistan as a distinct third category. As per the Transgender Person

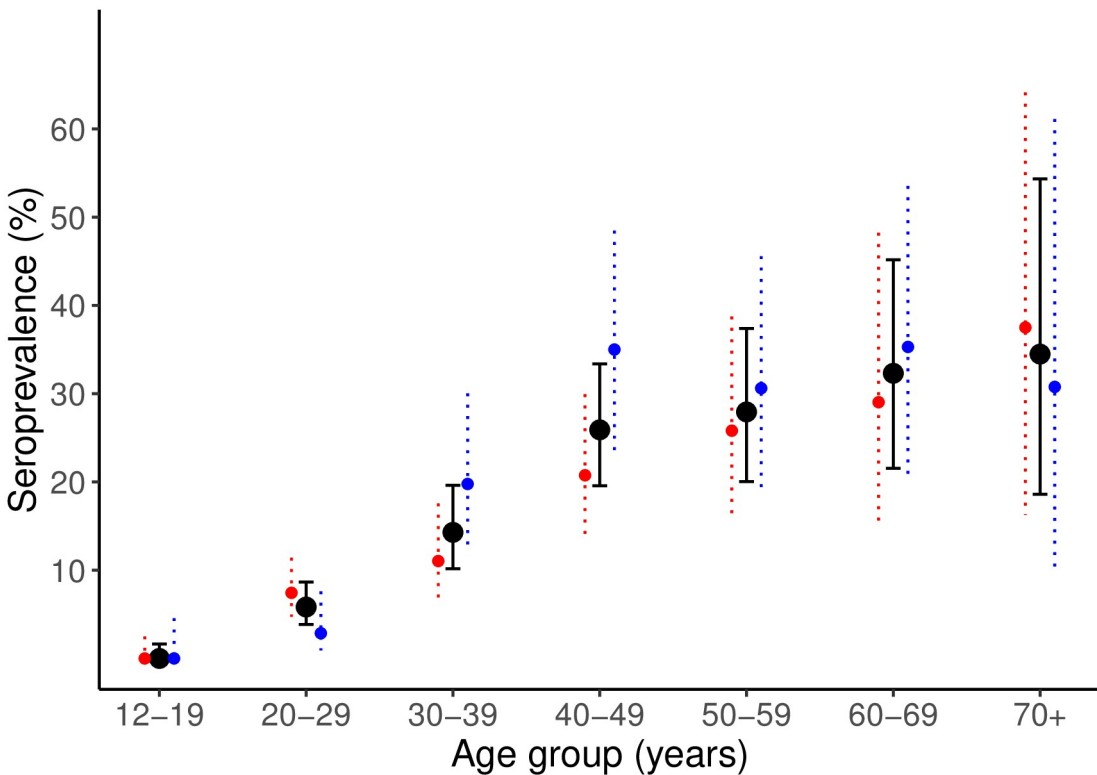

**Fig 2. Seroprevalence of HCV by 10-year age groups among the general population (black points), males (blue points), and females (red points) in Machar Colony, Karachi using data collected in March 2022.** Error bars represent 95% confidence intervals.

**Table 1. Characteristics of study participants in relation to HCV serostatus in Machar Colony, Karachi using questionnaire data and samples collected in March 2022.**

| Characteristic (number of responses) | | Total number (%) | Number (%) HCV seropositive |
|---|---|---|---|
| Gender (1300) | Female | 817 (62.8) | 89 (11.0) |
| | Male | 483 (37.2) | 73 (15.1) |
| Age (in years) (1300) | 12–17 | 165 (12.7) | 0 (0) |
| | 18–29 | 533 (41.0) | 24 (4.5) |
| | 30–39 | 231 (17.8) | 33 (14.2) _ |
| | 40–49 | 166 (12.8) | 43 (25.9) |
| | 50–59 | 111 (8.5) | 31 (27.9) |
| | >60 | 94 (7.2) | 31 (32.9) |
| Any therapeutic injection in previous 12 months (1221) | No | 655 (53.6) | 59 (9.1) |
| | Yes | 566 (46.4) | 96 (17.0) |
| Reported transfusion (1265) | No | 1173 (92.7) | 137 (11.7) |
| | Yes | 92 (7.3) | 22 (23.9) |
| Reported use of dentist (1232) | No | 1089 (88.4) | 123 (11.3) |
| | Yes | 143 (11.6) | 31 (21.7) |
| Work in fisheries sector (1295) | No | 1161 (89.0) | 134 (11.5) |
| | Yes | 134 (11.0) | 27 (20.2) |
| Knowledge about HCV (1300) | No | 911 (70.0) | 108 (11.9) |
| | Yes | 392 (30.0) | 54 (13.8) |

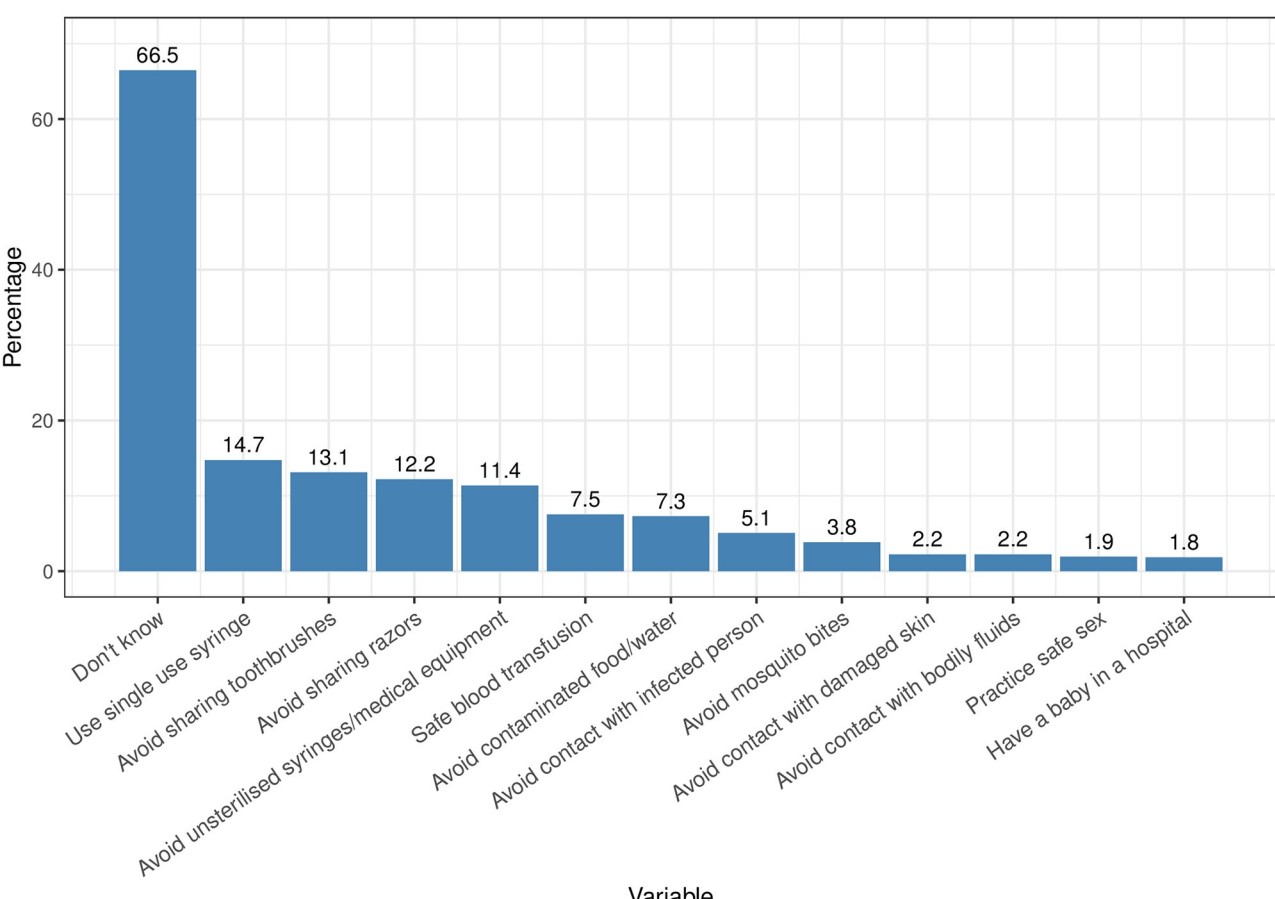

**Fig 3. Distribution of responses to the question "do you know any ways a person can avoid getting hepatitis C" among 1303 people interviewed in Machar Colony, Karachi in March 2022.**

(Protection of Rights) Act, 2018, this category was constructed to include those individuals who identify as either intersex, eunuch assigned male at birth, a transgender man, transgender woman, or any person whose gender identify differs from the social norms based on the sex they were assigned at birth [38]. Given this relatively small number, we excluded these individuals from the risk factor analysis so that the final data analyzed comprised 1300 observations (Table 1).

At least 30% of people were able to correctly identify at least one way to prevent HCV transmission. Fig 3 shows the breakdown of reported awareness in response to the question, "do you know any ways a person can avoid getting Hepatitis C?". Of those people reporting any answer, the most common (14.7% of people) was to use single use syringes. A total of 20% of people reported either that single use syringes should be used or to avoid unsterilised syringes and contaminated medical equipment. Fig 3 includes several responses of activities or exposures to avoid that are not considered to be risk factors for HCV infection, including avoiding contaminated food and water (7.3%), contact with infected people (5.1%), and mosquito bites (3.8%).

## 4.2. Predictors of HCV exposure

The random effect specification with the lowest predictive error was grid cell alone, with a DIC of 1159, compared to 1404 for the model with random effects at the household level alone and

**Table 2. Odds ratio estimates from multivariable logistic regression analysis of HCV seropositivity in Machar Colony, Karachi using test and questionnaire data collected in March 2022.**

| Predictor | Univariable Odds Ratio (95% CrI) | Multivariable Odds Ratio (95% CrI) | Posterior probability |
|---|---|---|---|
| Age | 1.07 (1.06–1.08) | 1.12 (1.09–1.15) | - |
| Age x age | - | 0.97 (0.96–0.98) | - |
| Female | 0.67 (0.47–0.95) | 0.89 (0.57–1.40) | - |
| Number of therapeutic injections in previous 12 months | 1.10 (1.05–1.16) | 1.07 (1.00–1.13) | 0.02 [a] |
| Reported transfusion | 2.41 (1.38–4.10) | 1.72 (0.90–3.21) | 0.05 [a] |
| Work in fisheries sector | 1.88 (1.14–3.05) | 1.51 (0.80–2.86) | 0.10 [a] |
| Awareness about HCV | 1.22 (0.83–1.79) | 0.93 (0.59–1.46) | 0.38 [b] |
| Reported use of dentist | 2.38 (1.47–3.83) | 0.94 (0.53–1.65) | 0.58 [a] |

[a] Probability odds ratio less than 1 for risk factors;

[b] probability odds ratio greater than 1 for expected protective effects

1272 for the model with random effects at the household and grid levels. Table 2 gives the observed relationships between HCV seropositivity and the included predictors in the full model. There was strong evidence (i.e., 95% credible interval for the odds ratio (OR) that does not include one) for a positive effect of increasing age. There was also strong evidence for a negative quadratic effect, supporting the existence of a plateau effect in HCV seropositivity with increasing age (Fig 2). There was no evidence of any difference in HCV serostatus based on male and female gender. There was good evidence for a relationship between HCV seropositivity and the number of therapeutic injections reported to have been received in the previous 12 months. The posterior probability for this estimate of 0.02 suggests that, based on the model specification and available data, we can be confident of a positive relationship between an increasing number of therapeutic injections and HCV seropositivity in this setting, or that there is low probability the OR for the relationship is less than one. A positive relationship between HCV seropositivity and a reported history of receiving a transfusion and working in the fisheries sector was also observed. The evidence for these associations was less strong than that for therapeutic injections, with 95% credible intervals for ORs crossing one and posterior probabilities that the odds ratio is less than one of 5% and 10%, respectively (Table 2 and Fig 3). There was no evidence for a relationship between HCV seropositivity and reported use of a dentist or our indicator of awareness about HCV. The observed odds ratios and 95% credible intervals from the regressions of variables with missing data (use of dentist, transfusion, work in fisheries and number of reported injections in past 12 months, Table 1) are shown in Table A in S4 File.

The observed odds ratios and 95% credible intervals from the regressions of variables with missing data (use of dentist, transfusion, work in fisheries and number of reported injections in past 12 months, Table 1) are shown in Table A in S4 File. Increasing age was positively associated with the number of injections received in the past 12 months, reports of receiving a transfusion, and ever using a dentist. There was limited evidence for an association between age and working in the fisheries sector. Females reported receiving a lower number of injections in the past 12 months than males and were much less likely to work in the fisheries sector. Females were more likely to report having received a transfusion.

## 4.3. Predictors of HCV viraemia

Table 3 shows HCV viraemic status by age and sex for the study population. Table 4 gives the results from the multivariable logistic regression of the relationship between HCV viraemia

**Table 3. Characteristics of study participants in relation to HCV viraemia in Machar Colony, Karachi using questionnaire data and samples collected in March 2022.**

| Characteristic | | Total number (%) | Number (%) HCV viraemic |
|---|---|---|---|
| Gender[1] | Female | 801 (62.9) | 21 (2.6) |
| | Male | 472 (37.1) | 22 (4.7) |
| Age (in years)[1] | 12–17 | 165 (13.0) | 0 (0) |
| | 18–29 | 534 (41.9) | 10 (1.9) |
| | 30–39 | 228 (17.9) | 10 (4.4) |
| | 40–49 | 158 (12.4) | 8 (5.1) |
| | 50–59 | 107 (8.4) | 9 (8.4) |
| | >60 | 84 (6.6) | 6 (7.1) |

[1] Excluding the 27 individuals who did not present to the MSF clinic for confirmatory testing

**Table 4. Odds ratio estimates from multivariable logistic regression analysis of HCV viraemia among 162 seropositive people in Machar Colony, Karachi using test and questionnaire data collected in March 2022.**

| Predictor | Univariable Odds Ratio (95% CrI) | Multivariable Odds Ratio (95% CrI) |
|---|---|---|
| Age | 0.90 (0.62–1.29) | 0.95 (0.63–1.42) |
| Female | 0.73 (0.35–1.51) | 0.94 (0.38–2.29) |
| No previous hepatitis C treatment reported | 29.6 (9.0–130.9) | 30.4 (9.12–132.5) |

and age, sex, and reported previous treatment among HCV seropositive people. There was no evidence of a relationship between age or sex and the odds that an individual was HCV viraemic. There was a very strong association between participant reported history of previous treatment and the odds that a seropositive individual was viraemic, with those seropositive people reporting no previous treatment having around 30 times the odds of viraemia compared to those reporting previous treatment. Based on this multivariable model, the probability that a seropositive individual reporting previous treatment was viraemic was 4.8% (95% CrI 1.0–11.2) while the probability among those reporting no previous treatment was 55.3% (95% CrI 43.8–66.6). Characteristics of seropositive people who did and did not attend the MSF hepatitis C clinic for confirmatory testing are shown in Table B in S4 File.

### 4.4. Spatial clustering in HCV seropositivity

We found significant RSA at the grid level in the null model (Moran's I statistic = 0.10, p = 0.001). The included predictors had no impact on grid level RSA in the full model (Moran's I statistic = 0.10, p = 0.001).

A cluster of significantly elevated grid-level seropositivity risk was detected in the southeastern part of Machar Colony with a relative risk (RR) of seropositivity inside the cluster compared to outside of 1.29 (p-value = 0.007). A cluster of significantly reduced grid-level risk was detected in the north-western part of Machar Colony, with a RR of 0.82 (p-value = 0.02) (Fig 4).

### 5. Discussion

The observed survey adjusted population-level HCV seroprevalence in Machar Colony of 13% is substantially higher than the national average for Pakistan of 4.8% and the average of 5% for

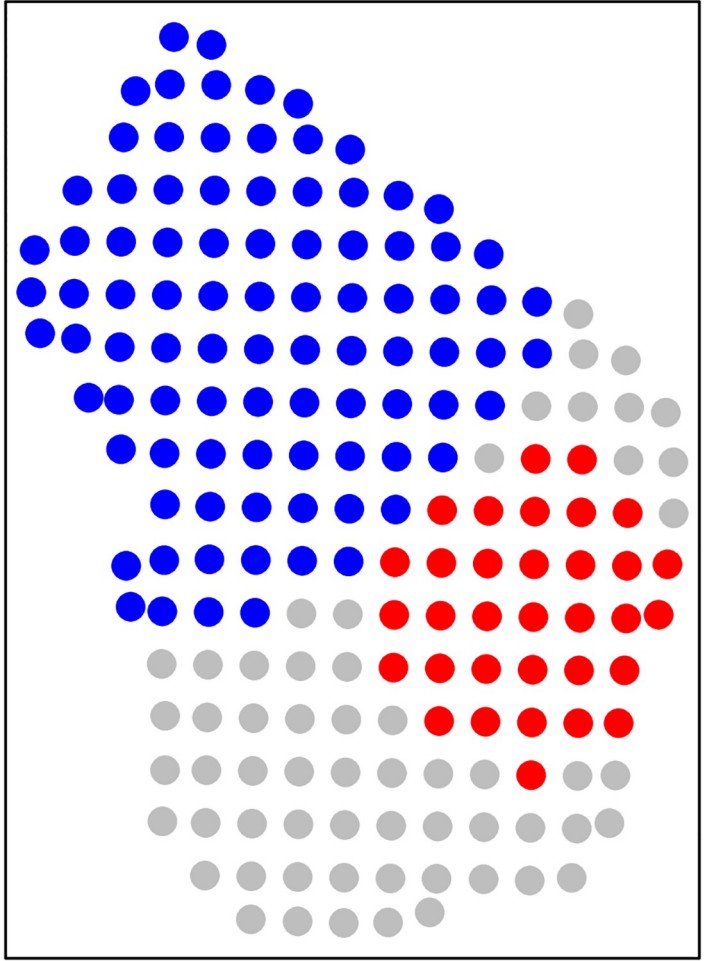

**Fig 4. Clusters of significantly elevated (red, relative risk (RR), 1.29, p-value = 0.007) and reduced (blue, RR = 0.82, p-value = 0.02) grid cell risk of HCV seropositivity in Machar Colony, Karachi using data collected in March 2022.** Grey grid cells were not found to be included in any clustering.

Sindh Province [6, 8]. We find evidence for an increasing number of therapeutic injections received in the past 12 months as a risk factor for HCV seropositivity, suggesting iatrogenic sources of infection as likely drivers of transmission among the general population in this setting. We found a low viraemic prevalence relative to seroprevalence, with around one third of people that were HCV seropositive in this community having viraemic infections.

Several studies have reported therapeutic injections as an important risk factor for HCV infection in Pakistan [39–42]. People in Pakistan receive an unusually high number of therapeutic injections, estimated at between 4 to 5 per person annually. An estimated 17 to 50% of these are given with reused syringes [43]. High injection use in the country is driven by high patient demand due to a very strong belief among patients in Pakistan of the efficacy of injections in providing faster and better relief than oral drugs [44–46]. We found a relatively lower number of reported annual injections in Machar Colony, with an average of 1.5. However, we observed a strong relationship between HCV seropositivity and the reported number of therapeutic injections received by participants in the past 12 months, suggesting exposure to HCV-contaminated needles and syringes is likely to be a source of HCV infection in this setting. There is an urgent need for further public health interventions to mitigate this. This could

include awareness drives for both healthcare providers and patients that are centered on education around harm reduction. It is notable that only 20% of people in this setting reported that HCV risks can be reduced through safe injection practices. In a study conducted to measure the level of awareness amongst patients suffering from hepatitis C in Karachi, 61% of patients thought that exposure to needles/syringes was the cause of their illness [47]. Of 300 patients attending family medicine clinics in Karachi, 94% men and 53% women were aware that hepatitis C can be transmitted through unsterilized needles and surgical instruments [48]. These previous reports suggest awareness about transmission of HCV through needle and syringe re-use is lower in Machar Colony, than in wider Karachi. Although legislation requiring the sale and use of safety syringes exists in Pakistan, control of HCV has been said to be a "distant dream" unless the Government can achieve better regulation of syringe and needle reuse in the medical sector [49].

Our results also highlight a positive relationship between HCV seropositivity and ever receiving a transfusion. We do not have data on when each participant's transfusion was received, but these findings strengthen previous evidence indicating that HCV transmission in Pakistan may be driven by healthcare procedures, with, blood transfusions as a predominant risk factor [2, 7, 10]. Although females were not at elevated risk of HCV seropositivity, we found that they were much more likely to have received a blood transfusion than males (Table A in S4 File), as is expected due to the risks associated with childbirth [50]. Other studies in Pakistan report an association between HCV seropositivity and the number of gestations [42]. Pakistan has a highly fragmented and demand-driven blood transfusion system with poorly regulated transfusion practices. This includes inadequate initial screening of donors for high-risk behaviors due to insufficient training of blood bank staff coupled with a lack of universal quality assured serological screening procedures. Additionally, infection control for blood transfusion has been poor, especially in Karachi, where practices depart from infection control standards, especially in reference to waste management, disposal and staff safety [51, 52].

We included working in the fisheries sector in our risk factor evaluation due to anecdotal reports from Machar Colony suggesting practices such as sharing knives for gutting fish and deveining shrimp are common. Such practices may pose a specific occupational risk for HCV infection among this group through exposure to contaminated sharps. However, the strength of evidence for the positive association we observed between HCV infection and working in the fisheries sector was weak and we were able to provide limited control for possible confounders. For example, others have found high HCV infection prevalence among fishermen which was linked to high levels of intravenous drug use among this group [53]. Machar Colony is a socially conservative community and we did not collect data on this risk factor, or other risk factors known to be important for HCV transmission in other settings but which would have had the potential to cause offence and impact on survey engagement in this community. Education on the risks of HCV and promotion of occupational safety around handling of knifes and other sharps among people working in the fisheries sector is likely to be beneficial as part of wider health promotion messaging in this community.

Our data suggest a significantly higher risk for HCV infection in the southeastern part of Machar Colony. It is notable that work in the fisheries sector is particularly common in this part of Machar Colony, and many of the fish processing factories are in this area. However, the inclusion of work in the fisheries sector, or any other variable, in the model did not explain the spatial autocorrelation in residual HCV infection risk observed. Further work is required to understand why the southern part of Machar Colony is at elevated risk for HCV infection, which could further support the targeting of future interventions to particularly high-risk groups. Demonstration of spatial clustering of infection in this relatively small urban area

highlights that important heterogeneities in HCV infection risk can exist in small areas, including within a single slum settlement.

It is generally thought that between 20 and 40% of people with an acute HCV infection will spontaneously clear the virus [54, 55]. These people will remain seropositive for life, resulting in an expected viraemic ratio among an untreated population of between 60 and 80%. We found a viraemic ratio in Machar Colony of 32%. It is likely that this lower-than-expected viraemic ratio is due, at least in part, to the availability of convenient and free of cost HCV screening and treatment services offered consecutively by two MSF clinics in Machar Colony since 2015. This hypothesis is supported by the finding that 44% of seropositive people reported receiving treatment for CHC and that almost 80% of people reporting previous HCV testing reported that this had been performed in Machar Colony. Unsurprisingly, we found a very strong association between HCV viraemia and reported history of receiving treatment for hepatitis C in the past among seropositive people. In people reporting no previous treatment for hepatitis C, the viraemic ratio was estimated to be 55%, further supporting the hypothesis that the availability of HCV treatment has substantially reduced the population-level prevalence of HCV viraemia. However, further work is required to fully explore the impacts of the free of cost and easily accessible hepatitis C screening and treatment services in reducing the population-level prevalence of HCV viraemic infection in this setting, including control for migration in and out of the area and differential losses due to higher mortality rates among HCV infected people than uninfected people.

This study has several limitations. Given the context of the study setting as mentioned above, we excluded some socially sensitive risk factors from our questionnaire, which meant we could not explore risk factors such as drug use in Machar Colony. Our sample was skewed towards a higher proportion of women compared to men. This is due to field activities conducted in daylight hours for safety reasons. As expected, working age males were often away from the household at the time of sampling, with many males in this community also working on commercial fisheries and spending long periods at sea. To rectify the gender imbalance, mitigation strategies were deployed to improve the recruitment rate among men, such as offering screening before work, on weekends, and revisiting households to offer screening.

We also had moderate levels of missingness in predictors of HCV seropositivity. We do not expect stigma around the characteristics included as predictors (i.e., medical injections, dentist use, transfusions in this setting) and therefore considered it unlikely that these data were missing not at random. We therefore imputed the missing values. It is important to note however that there may have been non-random missingness in screening test data that may have impacted upon our findings in an unknown way. For example, if the people who tended to be away from home (and were therefore not sampled) were more likely to be HCV seropositive, as was observed among men engaged in commercial fishing in other settings [53], we may have underestimated both the seroprevalence and viraemic prevalence in Machar Colony. A further limitation is that almost 20% of seropositive people did not attend the MSF clinic after a positive screening test, despite follow ups by telephone and in person to re-explain the need for confirmatory testing. This reflects the wider challenges of linking people with positive screening results to care in this setting.

## 6. Conclusion

This study provides a scientific basis upon which to adapt and design healthcare services, as well as future interventions that are intended to reduce the prevalence and incidence of HCV infection in this highly marginalized community. With an improved understanding of the seroprevalence, viraemic prevalence and risk factors for infection coupled with identification

of spatial heterogeneity of HCV, more intensive screening in particularly high-risk areas and effective planning of treatment and prevention services can be designed. There is a particular need to implement targeted programs catered towards harm reduction strategies, which will aim to create awareness amongst informal and formal healthcare providers and the general population regarding risk factors for HCV, particularly around the reuse of injection equipment. This will not only bring benefits to this highly disadvantaged community but will also support Pakistan's National HCV elimination goals.

## Supporting information

**S1 File. Study data.**
(CSV)

**S2 File. Data dictionary.**
(RTF)

**S3 File. Survey questionnaire.**
(PDF)

**S4 File. Supplementary information.**
(DOCX)

## Acknowledgments

We are grateful to Dr. Rabia Maniar at IRD, Waqas Ahmed, Natalie Van Gijsel and Dr. Gul Khalid at MSF for their support during the Project.

## Author Contributions

**Conceptualization:** Munazza Mansoor, William A. de Glanville, Khawar Aslam, Aneeta Pasha.

**Data curation:** Munazza Mansoor, Ridwa Alam.

**Formal analysis:** William A. de Glanville.

**Funding acquisition:** Aneeta Pasha.

**Investigation:** Munazza Mansoor, Khawar Aslam, Mubashir Ahmed, Aneeta Pasha.

**Methodology:** Munazza Mansoor, William A. de Glanville, Khawar Aslam.

**Project administration:** Munazza Mansoor, Ridwa Alam.

**Resources:** Munazza Mansoor, Ridwa Alam.

**Software:** Ridwa Alam.

**Supervision:** Munazza Mansoor, Ridwa Alam, Khawar Aslam, Petros Isaakidis, Aneeta Pasha.

**Validation:** William A. de Glanville, Ridwa Alam.

**Visualization:** William A. de Glanville, Ridwa Alam.

**Writing – original draft:** Munazza Mansoor, William A. de Glanville, Ridwa Alam, Aneeta Pasha.

**Writing – review & editing:** Munazza Mansoor, William A. de Glanville, Khawar Aslam, Mubashir Ahmed, Petros Isaakidis, Aneeta Pasha.

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
