## [Decision Letter · Decision Letter 0]

5 Apr 2023

PGPH-D-23-00220

Prevalence and risk factors for hepatitis C virus infection in an informal settlement in Karachi, Pakistan

Dear Dr. Mansoor,

Thank you for submitting your manuscript to PLOS Global Public Health. After careful consideration, we feel that it has merit but does not fully meet PLOS Global Public Health’s publication criteria as it currently stands. Therefore, we invite you to submit a revised version of the manuscript that addresses the points raised during the review process.

Two independent reviewers have assessed the manuscript. Although they found the work interesting, they have raised some concerns that should be carefully addressed. We consider this a major revision. Please note that major revisions may be subject to re-review. Please pay special attention to comments regarding the statistical analysis.

We look forward to receiving your revised manuscript.

Kind regards,

Chrispin Chaguza, Ph.D

Academic Editor

Journal Requirements:

1. Please send a completed 'Competing Interests' statement, including any COIs declared by your co-authors. If you have no competing interests to declare, please state "The authors have declared that no competing interests exist". Otherwise please declare all competing interests beginning with the statement "I have read the journal's policy and the authors of this manuscript have the following competing interests:"

2. Please provide a/amend your detailed Financial Disclosure statement. This is published with the article. It must therefore be completed in full sentences and contain the exact wording you wish to be published.

a. Please clarify all sources of funding (financial or material support) for your study. List the grants (with grant number) or organizations (with url) that supported your study, including funding received from your institution. 

b. State the initials, alongside each funding source, of each author to receive each grant.

c. State what role the funders took in the study. If the funders had no role in your study, please state: “The funders had no role in study design, data collection and analysis, decision to publish, or preparation of the manuscript.”

d. If any authors received a salary from any of your funders, please state which authors and which funders.

If you did not receive any funding for this study, please simply state: “The authors received no specific funding for this work.”"

3. Some material included in your submission may be copyrighted. According to PLOS’s copyright policy, authors who use figures or other material (e.g., graphics, clipart, maps) from another author or copyright holder must demonstrate or obtain permission to publish this material under the Creative Commons Attribution 4.0 International (CC BY 4.0) License used by PLOS journals. Please closely review the details of PLOS’s copyright requirements here: PLOS Licenses and Copyright. If you need to request permissions from a copyright holder, you may use PLOS's Copyright Content Permission form.

Potential Copyright Issues:

Figure 1: please (a) provide a direct link to the base layer of the map (i.e., the country or region border shape) and ensure this is also included in the figure legend; and (b) provide a link to the terms of use / license information for the base layer image or shapefile. We cannot publish proprietary or copyrighted maps (e.g. Google Maps, Mapquest) and the terms of use for your map base layer must be compatible with our CC-BY 4.0 license. 

4. We have noticed that you have a list of Supporting Information legends in your manuscript. However, there are no corresponding files uploaded to the submission. Please upload them as separate files with the item type 'Supporting Information'. 

Reviewers' comments:

Reviewer's Responses to Questions

**Comments to the Author**

1. Does this manuscript meet PLOS Global Public Health’s publication criteria? Is the manuscript technically sound, and do the data support the conclusions? The manuscript must describe methodologically and ethically rigorous research with conclusions that are appropriately drawn based on the data presented.

Reviewer #1: Yes

Reviewer #2: Yes

2. Has the statistical analysis been performed appropriately and rigorously?

Reviewer #1: Yes

Reviewer #2: No

3. Have the authors made all data underlying the findings in their manuscript fully available (please refer to the Data Availability Statement at the start of the manuscript PDF file)?

Reviewer #1: Yes

Reviewer #2: Yes

4. Is the manuscript presented in an intelligible fashion and written in standard English?

Reviewer #1: Yes

Reviewer #2: Yes

5. Review Comments to the Author

Reviewer #1: The authors discuss an important topic, touching on the prevalence and risk factors for hepatitis C virus infection in Pakistan. The manuscript is well written and structured. Below are some issues the authors might want to address:

Statement on page 189 is not clear (2.5 Data Collection) “Participating household members conducted an individual questionnaire with close-ended 190 questions focusing on behaviors and exposures that have been shown to be important as risk 8 191 factors for HCV infection in other settings in Pakistan [9, 10, 26, 27].” I guess the point is that the questionnaire was administered to the participants.

Number of injections is one of the important independent variables in the analysis. The distribution of the number of injections received is likely to be right-skewed. Did the authors check the distribution of the number of injections to decide if treating it as categorical would make more sense?

The authors could first explore the missing data mechanism (MCAR, MAR, MNAR etc) before including models that augment or account for the missing data. They could check, for example, whether the probability that the data was missing, depends on age and sex. What has been presented in Table 3 are the results from the models for the missing values if I am not mistaken.

Before reporting results from the full model in Table 2, did the authors investigate how the predictors were associated with the outcome on their own? The crude and adjusted odds ratios could then be reported side-by-side. It would make sense to clearly state that these were adjusted odds ratios. Is there a why reason posterior probabilities are not reported for age in Table 2?

Figure 3 is not clear. Please consider increasing the resolution and making the labels on the x-axis clearer.

Reviewer #2: This paper presents the results of a serosurvey of hepatitis C conducted in a slum in Karachi.

The sampling framework is well described and appropriate.

Comments:

1. I would not consider the inclusion of the finding of the association with working in fisheries in the abstract since this association has wide confidence intervals substantially overlapping a null effect.

2. "The seroprevalence was more than double the reported seroprevalence in the Sindh Province."

Should include "previously reported"

3. This section is redundant and can be removed: "People who are HCV seropositive may fall into three categories: 1. Infected with HCV in the past but naturally cleared the infection; 2. Infected in the past and treated and cured; 3. Currently HCV infected."

4. "Individuals found to have hepatitis C infection were managed as patients following standard MSF clinical protocols." It would be informative to briefly describe what therapy was provided.

5. I am uncertain how the Bernoulli trial method for estimating viraemic prevalence has contributed to the validity of the viraemic prevalence estimates. A multiple imputation method would be more likely to be informative. An analysis of the known characteristics of participants who did, and did not attend for confirmatory testing will be useful to ascertain whether bias is likely to affect viraemic estimates.

6. A random effects variable was included at the grid level but not household level? A multilevel model should account for household level clustering, since transmission within the household might be expected.

7. The imputation of missing data for the multivariable logistic regression has not been fully described. What assumptions were made about missingness and how much missing data was imputed? Please describe this in a supplementary appendix.

8. The selection process as described is a two-stage sampling method with grid- and household level sampling. Prevalence should be adjusted by sampling weights accounting for the two-stage process rather than being treated as a simple random sampling in the analysis. I cannot see any mention of survey weights. These should be incorporated into all models including prevalence estimation and regression.

9. The table 1 baseline characteristics should include age summary statistics.

6. PLOS authors have the option to publish the peer review history of their article (what does this mean?). If published, this will include your full peer review and any attached files.

**Do you want your identity to be public for this peer review?** For information about this choice, including consent withdrawal, please see our Privacy Policy.

Reviewer #1: No

Reviewer #2: No

---

## [Decision Letter · Decision Letter 1]

10 Jul 2023

PGPH-D-23-00220R1

Prevalence and risk factors for hepatitis C virus infection in an informal settlement in Karachi, Pakistan

Dear Dr. Mansoor,

Thank you for submitting your manuscript to PLOS Global Public Health. After careful consideration, we feel that it has merit but does not fully meet PLOS Global Public Health’s publication criteria as it currently stands. Therefore, we invite you to submit a revised version of the manuscript that addresses the points raised during the review process.

The reviewers' comments are positive, and we expect to accept this manuscript soon for publication. However, before formally accepting the manuscript, we would like you to address a few minor reviewer comments. The reviewers' comments seem straightforward and can be addressed in a few days.

We look forward to receiving your revised manuscript.

Kind regards,

Chrispin Chaguza, Ph.D

Academic Editor

Journal Requirements:

b. If any authors received a salary from any of your funders, please state which authors and which funders.

Additional Editor Comments (if provided):

Reviewers' comments:

Reviewer's Responses to Questions

**Comments to the Author**

1. If the authors have adequately addressed your comments raised in a previous round of review and you feel that this manuscript is now acceptable for publication, you may indicate that here to bypass the “Comments to the Author” section, enter your conflict of interest statement in the “Confidential to Editor” section, and submit your "Accept" recommendation.

Reviewer #1: All comments have been addressed

Reviewer #2: (No Response)

2. Does this manuscript meet PLOS Global Public Health’s publication criteria? Is the manuscript technically sound, and do the data support the conclusions? The manuscript must describe methodologically and ethically rigorous research with conclusions that are appropriately drawn based on the data presented.

Reviewer #1: Yes

Reviewer #2: Yes

3. Has the statistical analysis been performed appropriately and rigorously?

Reviewer #1: Yes

Reviewer #2: Yes

4. Have the authors made all data underlying the findings in their manuscript fully available (please refer to the Data Availability Statement at the start of the manuscript PDF file)?

Reviewer #1: Yes

Reviewer #2: Yes

5. Is the manuscript presented in an intelligible fashion and written in standard English?

Reviewer #1: Yes

Reviewer #2: Yes

6. Review Comments to the Author

Reviewer #1: All the comments I made have been addressed. Table 1 could be formatted better. For example, the authors present Age (in years) (1300 (100%)) to show the number of responses and percentage of the total. The (100%) could be removed and a footer could be used to indicate the N for variables that had missing responses from some participants.

Reviewer #2: The changes made in this revised article have improved the manuscript.

I have a few additional comments:

1. I would update table 1 to also include viraemic prevalence (with the proportion and CIs adjusted for available data).

2. Table 3 presents evidence of the association between indicator variables of sex and age vs other regression variables. As it does not report an relationship with HCV prevalence I would remove this table to the appendix, as it is may be easily misinterpreted as referring to HCV.

3. The authors have not reported participant characteristics data stratified by present/missing HCV RNA result in a supplementary table as requested in the first review. This remains an important step to identify response bias and I recommend it.

4. Could figure 4 be accompanied with any additional spatial information/satellite imaging which might inform the observed clustering? Could figure 4 show any relevant spatial clustering statistics.

5. Could figure 2 also be stratified by sex (eg. male/female estimates for each age group)? Also the x-axis is categorical it should therefore include the proper categorical labelling eg. "15-19" and not "15". Better yet, use of a spline for age with 95% CI or CrI would show the continuous relationship overall, and stratified by sex.

7. PLOS authors have the option to publish the peer review history of their article (what does this mean?). If published, this will include your full peer review and any attached files.

**Do you want your identity to be public for this peer review?** For information about this choice, including consent withdrawal, please see our Privacy Policy.

Reviewer #1: No

Reviewer #2: No

---

## [Editor Report · Decision Letter 2]

25 Aug 2023

Prevalence and risk factors for hepatitis C virus infection in an informal settlement in Karachi, Pakistan

PGPH-D-23-00220R2

Dear Mansoor,

We are pleased to inform you that your manuscript 'Prevalence and risk factors for hepatitis C virus infection in an informal settlement in Karachi, Pakistan' has been provisionally accepted for publication in PLOS Global Public Health.

Please note, one change needs to be made. The table numbers should be revised since Table 3 was moved to the Supplementary Material. Currently, there is no Table 3 in the main text, so Table 4 should be named Table 3, Table 5 as Table 4, and so on.

Best regards,

Chrispin Chaguza, Ph.D

Academic Editor
